# Impact of Social Isolation Due to COVID-19 on Daily Life Activities and Independence of People over 65: A Cross-Sectional Study

**DOI:** 10.3390/ijerph20054177

**Published:** 2023-02-26

**Authors:** María Laura Frutos, David Pérez Cruzado, Dianna Lunsford, Santiago García Orza, Raquel Cantero-Téllez

**Affiliations:** 1Facultad Ciencias de la Salud, Universidad Católica de Córdoba, Córdoba 5000, Argentina; 2Facultad de Ciencias de la Salud, Departamento de Fisioterapia, Universidad de Málaga, 29071 Málaga, Spain; 3Health Sciences Faculty, Occupational Therapy Department, Gannon University, Ruskin, FL 33573, USA; 4Hospital Comarcal de la Axarquía, 29700 Vélez-Málaga, Spain; 5FE-17 HandResearchGroup IBIMA, 29590 Málaga, Spain

**Keywords:** COVID-19, elderly, independence, activities of daily living

## Abstract

The mandatory confinement caused by the COVID-19 pandemic has significantly affected the older adult population. The main objective of this study is to assess independence in basic activities of daily living (BADL) and instrumental activities of daily living (IADDL) of people over 65 years of age during social, preventive, and compulsory isolation due to COVID-19, identifying and quantifying the activities of personal independence that present difficulties in their execution. Design: A cross-sectional study. Settings: Private’s health insurance Hospital, Córdoba, Argentina. Participants: A total of 193 participants with mean age of 76.56 years (121 women and 72 men) who met inclusion criteria were included in the study. Interventions: A personal interview was conducted between July and December 2020. Sociodemographic data were collected, and perceived independence was assessed. Outcomes measures: The Barthel index and the Lawton and Brody scale were used to assess independence of basic and instrumental activities of daily living. Results: Minimal limitations were noted with function. The activities that represented the greatest difficulties were going up and down stairs (22%) and moving around (18%), and the greatest difficulties in instrumental activities of daily living were shopping (22%) and preparing food (15%). Conclusions: COVID-19 has caused isolation, leading to functional limitations for many, especially older adults. Perceived declines in function and mobility may lead to decreased independence and safety for the older adult; therefore, preventative planning and programming should be considered.

## 1. Introduction

The increase in life expectancy in the aging of the population is a reality that can be analyzed from different approaches. The ability to care for oneself or maintain independence undergoes transformations later in life and “brings with it a gradual loss of physical and mental capacities, which increases the probability of losing autonomy to carry out basic activities” [1]. There are different strategies for maintaining the well-being of older adults; these include using technology to ensure social connections, pursuing outdoor activities, and incorporating daily structure. However, these improvements on the quality of life of older people have been affected by the pandemic because not all older people have access to new technologies that grew exponentially in the period of mandatory closure due to COVID-19. This fact may affect the quality of life of these people beyond the pandemic [1,2].

The Framework for the Practice of Occupational Therapy [2] defines activities of daily living (ADL) as those related to home maintenance, meal preparation, and interactions with the community, among the most important. Similarly, in the International Classification of Functioning, Disability and Health (ICF), self-care activities and domestic life are mentioned as categories of analysis within the concept of health [3].

Although it is currently unclear what the full extent of the effects of this pandemic will be, its negative impact on psychological well-being has become very evident. The disruption of workplaces, exercise routines, and widely imposed social isolation are all likely to have a large effect on the well-being of the population going forward. While there will not be a group of the population untouched by this crisis, the elderly population is likely to face the worst effects. Initial reports have shown that ~80% of the deaths due to COVID-19 occur in those over the age of 65. Despite the fact that studies prior to the pandemic caused by COVID-19 conclude that approximately 80% of older adults are independent in carrying out activities of daily living [4,5], the changes caused by the lack of mobility and social isolation during the last year and a half may have caused an alteration in trend for this older adult population. The World Health Organization (WHO) classified COVID-19 as a pandemic on 11 March 2020 (World Health Organization, 2020), caused by a respiratory virus called severe acute respiratory syndrome coronavirus 2 (SARS-CoV-2), first described in Wuhan, China, in December 2019 [6]. The first case of COVID-19 in Argentina was confirmed on 3 March 2020 [7]. The pandemic caused by COVID-19 has caused fear and suffering in the general population, but it is the elderly who have suffered the highest rate of illness and mortality, reaching five times the world average [8]. The virus has not only endangered the lives and safety of these elderly people, but social isolation also often results in loneliness, which is a factor significantly associated with depression in elderly adults. Loneliness, isolation, and depression have all been shown to predict worse disease outcomes in older populations. In addition, restrictions on freedom of movement and physical distancing have caused a decrease in essential care for this population, thus putting their social and economic well-being at risk [9].

Physical activity has a positive impact on health and quality of life, reducing the risk of functional and cognitive impairment, falls, depression, disability, risk of geriatric syndromes, hospitalization rates, and, consecutively, mortality in older people [10]. In addition, positive effects of physical activity have been described as impacting balance, strength, mobility, and activities of daily living in older people [11]. One of the first measures adopted by the WHO was to warn about the importance of social isolation to prevent a rapid spread of the disease in the population and avoid a collapse in health systems [12]. This preventive and compulsory social isolation caused a reduction in the level of physical activity that contributed to the development of different health problems, especially in older adults [6]. The negative effects of reduced physical activity due to mobility restriction and social isolation caused by COVID-19 have been demonstrated in various studies [13], but there is no consensus regarding the most appropriate type of complementary physical activity that could have been carried out in this population nor if said activity would have had positive effects for independence in the daily activities of the various study groups. Isolation limits group activities, workshops, and participation in associations or social gatherings, thus affecting cognitive levels during this period of isolation [14]. In relation to this, the emotional and mood state could have interfered with the level of independence when carrying out basic and instrumental activities of daily life [15]. Various studies have evaluated the effects of “confinement” on existing habits and pathologies. Most of these studies focused on the psychological aspects and psychosocial dynamics that derive from the situation itself [16,17,18], but the impact that social isolation and inactivity have had on the daily activities of the older adult population have not been studied in depth.

The objective of this study is to assess the independence in the basic activities of Daily living (BADL) and instrumental activities of daily living (IADL) of people over 65 years of age during social, preventive, and mandatory isolation due to COVID-19, identifying and quantifying the activities of personal independence that present difficulties in their execution.

## 2. Materials and Methods

### 2.1. Study Design

A cross-sectional observational study was carried out 6 months after the mandatory isolation period caused by COVID-19 began. The study was approved by the Institutional Ethic Committee of health investigation (CIEIS) Private University Hospital (Córdoba, Argentina) with number (HP 4-327). The ethical principles expressed in the Declaration of Helsinki [19] for medical research on human beings were respected at all times, and the data collected was safeguarded as established by Law No. 25.326 on “Protection of personal data” [20].

### 2.2. Participants

Participants older than 65 years, affiliated with Hospital Privado’s health insurance, who had access to the telephone, and who gave their informed consent to participate in the study were recruited for the study. Participants were excluded if they were institutionalized, had no records in their electronic medical records (EMR) in the last 6 (six) months, or those with cognitive impairment recorded in the EMRs. The sample number was estimated with the online software Raosoft^®^ [21] with a confidence of 95% and an error of no more than 7%; it yielded n = 191.

### 2.3. Outcome Variables

The Barthel index (or Barthel scale) is an instrument used for the functional assessment of a patient to determine the independence level in carrying out activities such as feeding, bathing, toileting, control of bowel and bladder, dressing, passing from bed to wheelchair, using a wheelchair/walking, and climbing stairs [22]. This scale consists of 10 items scored as 0 (unable), 5 (requires assistance), and 10 (independent), with a total sum ranging from 0 to 100. Additional assessment of the patient demonstrating an increase in total score indicates increasing levels of independence. The Barthel index (BI) has been recommended for the functional assessment of older people and was found to be reliable when administered by face-to-face interview and by telephone (ICC 0.89) [22].

The instrumental activity daily life (IADL) scale, developed by Lawton and Brody in 1969 to assess the more complex ADLs necessary for living in the community [23], was used to measure disability levels and assess parameters in community-dwelling older adults. This scale comprises eight items, including the ability to use a telephone, shopping, food preparation, housekeeping, laundry, use of public transportation, managing self-medication, and handling finances. Responses to each of the eight items on the scale are scored as 0 (cannot perform or can partially perform) or 1 (can perform). The total score ranges from 0 (low-functioning, dependent) to 8 (high-functioning, independent). Each ability measured by the scale relies on either cognitive or physical function though all require some degree of both. The scale can be administered with a written questionnaire or by interview [23].

### 2.4. Intervention

From the list of 6427 EMRs of those 65 years and older, 6271 were selected and randomized until reaching 400 participants to include in the study.

The 400 EMRs initially selected were read to identify exclusion criteria. Excluded patients included five institutionalized, four patients without registration in their EMR in the last six months, nine patients with cognitive impairment, and two deceased patients (n = 20). (Figure 1). A total of 380 patients met the established inclusion criteria.

Initially, the patients were contacted if an email was provided in the EMRs (n = 255). Reasons for the contact and the research were explained. If they agreed to participate, researchers requested that the participant reply to the email with date and appropriate time to conduct the telephone interview (within a time range of 2:00 p.m. to 8:00 p.m.). The written informed consent for telephone interview and the two evaluation scales (Barthel and Lawton and Brody) were attached to the email. A total of 255 emails were sent, of which 89 agreed to participate. One participant responded, but she did not wish to participate in the study. Of the 165 emails that were not answered, 92 participants were recruited via phone and agreed to participate in the study. An additional 12 participants who met the inclusion criteria were recruited from a randomized list from the rehabilitation service of the hospital. A total of 193 participants were included in this study.

### 2.5. Statistical Analysis

An ad hoc data matrix was designed including the following variables: age, sex, educational level, previous occupation, number of people at home, positive COVID, falls suffered during the period, and if they had any pathology that interferes with their ability to perform the BADL and IADL, Barthel index [22], and Lawton and Brody scale [23]. For the statistical analysis, the R MEDIC [24] software was used with an a priori significance level set at *p* ≤ 0.05. The relationship between the sociodemographic variables (except the number of people at home) the Barthel index and the Lawton and Brody scale was carried out using the Kruskal–Wallis test.

## 3. Results

A total of 193 participants were included in the study with mean age of 76.56 years (121 women and 72 men). Participant sociodemographic characteristics and the relationship between sociodemographic variables, the Barthel index, and the Lawton and Brody scale are displayed in Table 1.

Regarding the assessment through the Barthel index, 73% reported being independent in their BADL, reaching the maximum score of 100. Only 1% recorded lower percentages of independence. The greatest limitations reported by participants included going up and down the stairs (22%) and difficulty in moving around (18%). Significant differences were observed between the values of the Barthel index and the age ranges (*p*< 0.01) (Figure 2) and between the values of the Lawton and Brody scale and the age ranges (*p*< 0.001) (Figure 3)

The higher the age range, the lower the independence value. The lowest values of independence were recorded between the age ranges 90–94 years. For instrumental activities of daily living, shopping (22%) and food preparation (15%) were the activities presenting more difficulties in order of frequency (Table 2).

In the Lawton and Brody scale, 75% obtained the maximum score of 8 in IADL, and 5% obtained the minimum score demonstrating limitations in carrying out these activities. The values of the Barthel index and the Lawton and Brody scale do not present significant differences in relation to sex (*p* < 0.81*/p* < 0.31), education (*p* < 0.21/*p* < 0.23), or previous occupation in the Barthel index (*p* < 0.35), but significances were found with previous occupation (*p* < 0.03) in the Lawton and Brody scale (Figure 4). Considering the number of cohabitants at home, the greatest independence for IADLs was recorded in people whose households consisted of two people (−0.25) (*p*-value < 0.001). People who had COVID + diagnosis during this period did not present significant differences in the independence values in BADLs (Barthel) (*p* < 0.66) and IADLs (Lawton and Brody) (*p* < 0.31), respectively. Fifteen participants had suffered falls that required medical intervention, ranging from fractures or trauma to the upper limb (n = 11) to a fracture in the bones of the nose, in one case.

## 4. Discussion

Older adults are known to experience loneliness, age discrimination, and excessive worrying, and for this reason, we would think that they would experience greater negative outcomes and more dependency in their daily life related to the COVID-19 pandemic. However, this hypothesis is uniformly supported neither by the available literature nor by with the results of our study. The main objective of this research was to assess the independence in BADL and IADL of people over 65 years of age during social isolation due to COVID-19, identifying and quantifying the activities of personal independence that present difficulties in their execution. The literature review suggest that younger individuals were more concerned about the risks related to social isolation due to the mandatory isolation, but older adults reported a presence or worsening of psychological symptoms and greater loneliness because of pandemic-related social isolation. However, COVID-19 has not only impact caused changes in interpersonal relationships but also in the ability to move and physical activity, which has affected the elderly population above all [24,25].

The pandemic has caused disastrous socioeconomic consequences that could also lead to a great physical deterioration and independence [24]. In the present study, the mean age was 76.56 years, with slightly more women than men. In this preliminary study, we did not take into consideration the cognitive impairment of the participants, which could have influenced our results, as shown by previous research [25].^.^ People with cognitive difficulties have more problems in their functioning as independent people, hence the importance of developing future research focused on socio-demographic aspects such as level of study, lifestyle, or life history that allow us to determine to what extent each one of them can influence the independent activity of adults older than 65 years. Educational level, intelligence level, and active lifestyle are factors closely related to independence in old age, a fact that has been confirmed in both cross-sectional [26,27,28] and longitudinal studies [29,30]. In addition, social contacts and participation in social-type activities decrease the risk of cognitive decline in older people and therefore increase their independence in basic and instrumental activities of daily living [31,32]. These results are consistent with this current study, where 40% of our participants completed secondary education, and the majority considered themselves independent in basic (73%) and instrumental (75%) activities of daily living at the time of the study. The activities that presented the greatest difficulties for their execution were going up and down stairs (22%) and going shopping (22%). These data could be biased by the exceptional situation experienced in that period, which could have affected perceptions when answering the questionnaires, or by the impossibility of being able to carry out a face-to-face evaluation due to the specific characteristics of the period in which the investigation was carried out. Because various studies report that with increased age comes a decrease in the number of activities in which the older adult engages [33,34], a false hypothesis could be created to imply that isolation has not interfered significantly in the participation of instrumental activities of daily life in older adults. However, a retrospective and long-term evaluation would be necessary to obtain more conclusive results.

Regarding basic activities such as going up and down stairs, a higher percentage of interviewees found difficulty (22%), followed by mobility at home (18%) and transfer from chair to bed (7%). This may be due to the lack of physical activity caused by the restriction of mobility that causes a loss of balance, coordination, and strength. In the systematic review carried out by Yixiong Zhang et al. [35], it was concluded that physical activity and exercise have positive effects on older people in terms of muscle strengthening, mobility, balance, or reaction speed. Regarding the falls suffered at home during this period, this may be related to the lack of mobility and reduction of activities within the home of the elderly during said confinement; however, this percentage would be below the 30% of falls per year as reported by the Ministry of Social Development of Argentina [36] or the data reported in the National Survey on Quality of Life in Older Adults (NSQoLOA) [37] where it is stated that 32% of the older adults interviewed fell in the last two years, and of that total, more than half (56%) fell more than once.

Several variables that we did not take into consideration could be associated with the results reported in our study and those in the future. We considered level of studies, previous occupation, or education, but some changes in life habits due to COVID-19 could have interfered in the independence level of this population. For example, changes in sleep habits and sleep disturbances were reported to be a result of COVID-19 in previous studies [38,39,40,41], while some studies indicated that sleep issues were lower in older adults than in younger adults. Since the lack of rest could interfere with the subsequent development of daily activities, this variable should be considered in future studies [42,43]. Changes in daily routines and plans may not interfere with the ability to carry out daily activities, but changes in the performance of daily habits among older adults have been report in previous studies [44], and behavioral changes such as buying more food and water than usual, going out less frequently, reducing social contacts, and staying away from public places were noted in several studies [45,46]. However, these results cannot be compared with the results obtained in our study because we used the data from the participants when they were still in a situation of total isolation. For the same reason, we found contradictions in different studies that addressed physical activity of the elderly during the pandemic. It is important to consider the period in which the study is carried out since the results may vary depending on whether the analysis is carried out at the time of the confinement or when they were allowed to go outside. Since we did not carry out long-term follow-up, we cannot know if these variables of “lifestyle changes” could have caused alterations in the level of dependency of the study population.

However, it is necessary to consider the methodology that is carried out in numerous studies to obtain information about the effect that COVID-19 has had on the elderly. In our study, we conducted a direct telephone interview, which allowed us to interact with the person and even direct the interview or obtain subjective data that cannot be collected in another electronic format. This is one of the strengths of our study. Older adults may have little to no access to technology [47] such as a computer or a smartphone, which are often required to participate in web-based surveys. We must emphasize that in our study, mostly isolated individuals participated, and those who may present a higher level of dependence on basic and instrumental activities may be the most difficult to reach, particularly if they lack access to social media or maintain a minimal presence in public and community organizations. Even so, due to the period in which the interview was conducted, we did not obtain significant results in the dependence level of the participants.

Although we cannot establish an exact correlation since the period evaluated in our study does not exceed 6 months, it is a fact that is worth studying in greater depth to initiate future preventive plans. However, we should take into consideration that maybe the perception of independence that participants had at the time of the survey could be different at the end of the pandemic and should be considered in future studies. Doing so would support establishing prevention programs to improve and/or maintain the independence of daily activities and mobility. Future research should also focus on the long-term effects that the compulsory isolation period has had on the elderly not only in the aspect of independence in basic and instrumental activities of daily life but also in the cognitive capacity of older people.

## 5. Conclusions

COVID-19 has caused isolation, leading to functional limitations for many, especially older adults. The current study highlighted participants’ perceptions of decreased function in specific activities. Perceived declines in function and mobility may lead to limited independence and safety for older adults; therefore, preventative planning and programming should be considered.

## Figures and Tables

**Figure 1 ijerph-20-04177-f001:**
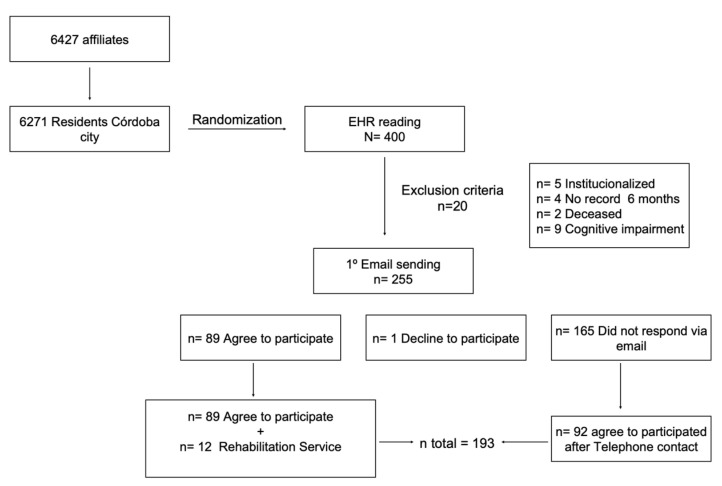
Flow Diagram.

**Figure 2 ijerph-20-04177-f002:**
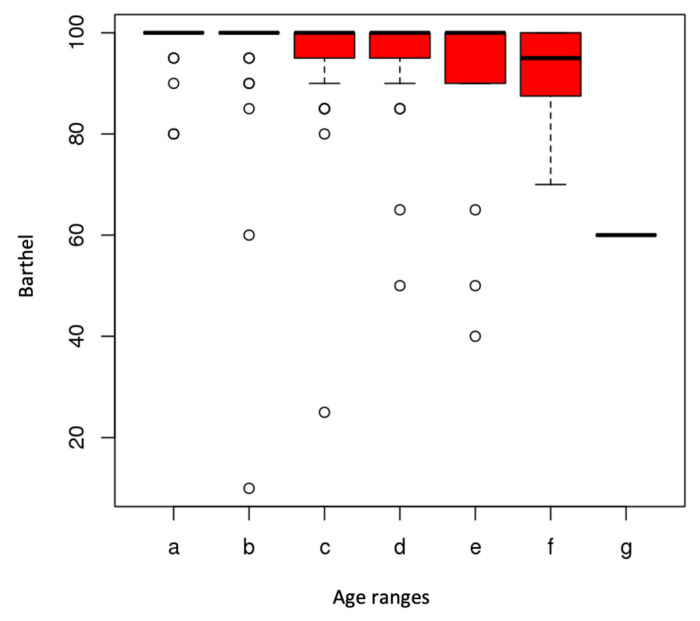
Relationship between Barthel and age ranges.

**Figure 3 ijerph-20-04177-f003:**
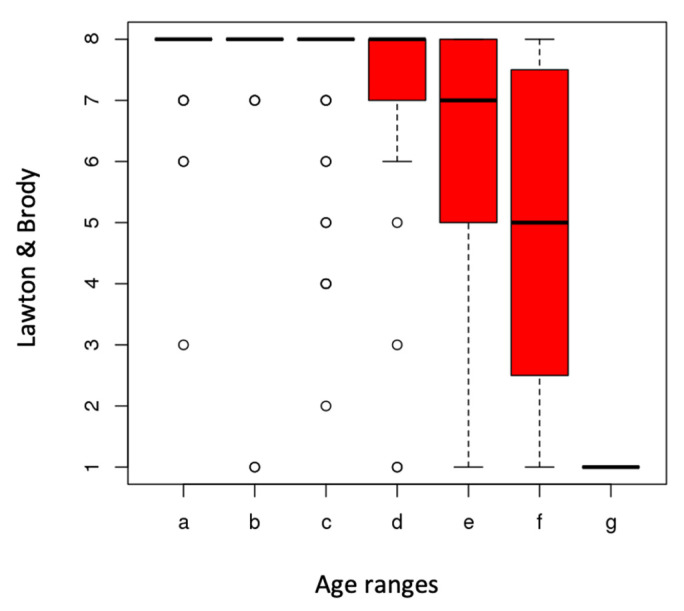
Relationship between Lawton and Brody and age ranges.

**Figure 4 ijerph-20-04177-f004:**
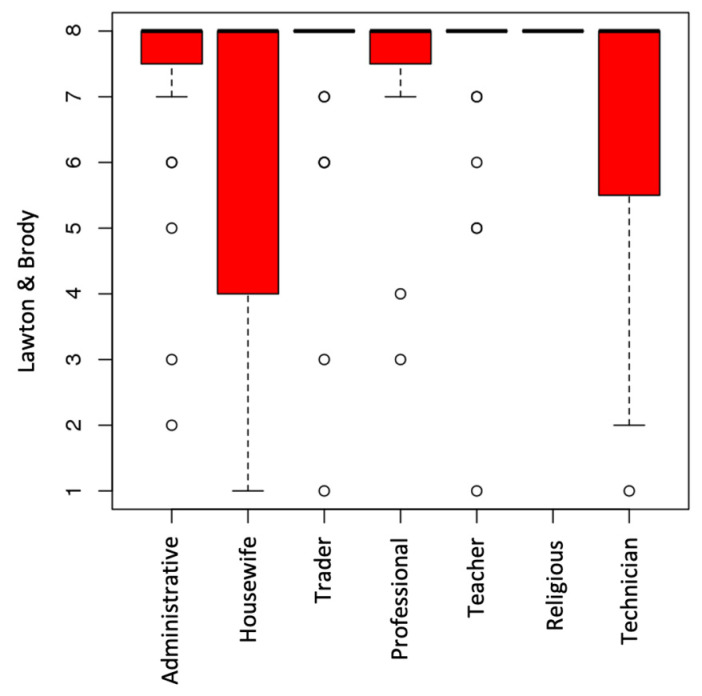
Relationship between Lawton and Brody and previous occupation.

**Table 1 ijerph-20-04177-t001:** Relationship between sociodemographic variables, the Barthel index, and the Lawton and Brody scale.

				Barthel		*p*-Value	Lawton/Brody		*p*-Value
		AF	%	Means	SD		Means	SD	
Age (range)						0.01			0.001
	65–69 (a)	38	20	98.42	4.81		7.68	0.93	
	70–74 (b)	40	21	95.63	15.53		7.58	1.55	
	75–89 (c)	52	27	95.96	11.25		7.4	1.36	
	80–84 (d)	40	21	95.38	9.96		7.15	1.78	
	85–89 (e)	15	8	88.33	19.88		6.07	2.76	
	90–94 (f)	7	4	91.43	11.8		4.86	3.02	
	95–99 (g)	1	1	60	N/A		1	N/A	
Sex						0.81			0.31
	Female	121	63	95.83	10.27		7.21	1.79	
	Male	72	37	94.44	15.33		7.22	1.81	
Education						0.21			0.23
	Incomplete primary	1	1	95	N/A		7	N/A	
	Complete primary	9	5	85.56	23.38		5.89	3.06	
	Incomplete secondary	9	5	94.44	11.3		7	2.35	
	Complete secondary	77	40	95.84	11.4		7.04	1.94	
	Incomplete tertiary	2	1	100	0		8	0	
	Complete tertiary	34	18	96.91	5.51		7.62	0.89	
	Incomplete university	13	7	85.77	26.91		6.62	2.6	
	Complete university	48	25	97.71	6.52		7.63	1.2	
Previous						0.35			0.03
occupation	Administrative	32	17	95.16	14.56		7.34	1.47	
	Housewife	30	15	90.5	17.49		6.1	2.8	
	Trade	39	20	95.9	7.94		7.49	1.43	
	Teacher	31	16	97.58	4.81		7.52	1.15	
	Professional	44	23	97.73	6.69		7.61	1.24	
	Religious	1	1	100	N/A		8	N/A	
	Technician	16	8	91.88	22.57		6.63	2.36	
Number of people									
at home						0.17			<<0.001
	1	46	24	97.61	4.56		7.72	0.72	
	2	117	61	97.09	6.7		7.45	1.42	
	3	20	10	84.5	26.3		5.3	3.13	
	4	8	4	81.88	28.53		5.63	2.92	
	5	1	1	100	N/A		6	N/A	
	8	1	1	100	N/A		8	N/A	

**Table 2 ijerph-20-04177-t002:** Difficulties in IADL (Instrumental Activities of Daily Living) in order of frequency.

Difficulty	AF	%
Shopping	43	22
Food preparation	28	15
Laundry	24	12
Housekeeping	20	10
Mode of transportation	15	8
Responsibility for own medications	12	6
Ability to handle finances	11	6
Ability to use telephone	0	0

## Data Availability

The research data information can be downloaded at: https://docs.google.com/spreadsheets/d/1T782VvWdWOoGautH5MVKdtOjjJshNWo6/edit#gid=1345954868 (accessed on 23 June 2022).

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
