# Peer review of "Impact of Social Isolation Due to COVID-19 on Daily Life Activities and Independence of People over 65: A Cross-Sectional Study"

_ijerph, 2023, doi:10.3390/ijerph20054177_

Round 1
Reviewer 1 Report
In the manuscript entitled ,,Impact of social isolation due to Covid-19 on daily life activities and independence of people over 65. A cross-sectional study’’ the authors showed participants perceptions in limitations due to the social isolation caused by COVID-19.
The topic is very interesting, and the authors showed some interesting results. However, I have some suggestion for revision:
1. In the Introduction section, page 2, line 70 you should cut off one of the duplicates ,,carried out’’.
2. In the Material and Methods section, the authors said ,,The sample number was estimated with the online software Raosoft ® (21) with a confidence of 95% and an error of no more than 7%, it yielded n= 191’’, but in whole papir you claim that a total number of participants is 193. How did the number of participants variate?
3. Do you have any example of questionnaire, or you have only interview by the telephone?
4. The figures are inadequately shown, there should be a legend under each figure or in the Supplementary Materials.
5. In the Results section, every Figure (1,2,3,4), every graph should be in English language.
6. Supplementary Materials should be sent in some other format, not Google Drive. I can’t get access to the Supplementary Materials.
7. In the Disscusion section, page 7, line 236 you should delete one of ‘or’.
Author Response
Comments and Suggestions for Authors
In the manuscript entitled ,,Impact of social isolation due to Covid-19 on daily life activities and independence of people over 65. A cross-sectional study’’ the authors showed participants perceptions in limitations due to the social isolation caused by COVID-19.
The topic is very interesting, and the authors showed some interesting results. However, I have some suggestion for revision:
- In the Introduction section, page 2, line 70 you should cut off one of the duplicates ,,carried out’’.
Authors respond: Thank you. Done.
- In the Material and Methods section, the authors said ,,The sample number was estimated with the online software Raosoft ® (21) with a confidence of 95% and an error of no more than 7%, it yielded n= 191’’, but in whole papir you claim that a total number of participants is 193. How did the number of participants variate?
Authors respond: The calculation estimate of the sample size indicates that the number of participants to obtain reliable results is 191. A higher percentage of participants should always be included initially because it is generally considered that 10% of the sample may be lost. In fact, our initial sample was well over 191 and as you can see in the flowchart, 255 participants were initially recruited, of which we were only able to interview 193.
- Do you have any example of questionnaire, or you have only interview by the telephone?
Authors respond: Yes, the questionnaires were computerized in the same system where the data was collected during the interview. We did it by phone due to the forced isolation due to COVID. All the data was collected in the questionnaires and then transferred to the computer for analysis. The questionnaires are deidentified and numbered, maintaining the anonymity of the participants. In the following link you can see the two scales that were passed to the participants by phone. https://drive.google.com/drive/folders/1EX556V6vjic3EmzbURrcPKhreUVKbxk5?usp=share_link
- The figures are inadequately shown, there should be a legend under each figure or in the Supplementary Materials.
Authors respond: Thank you. We have improved the figures. A legend is under each figure.
- In the Results section, every Figure (1,2,3,4), every graph should be in English language.
Authors respond: Thank you for the observation, changed.
- Supplementary Materials should be sent in some other format, not Google Drive. I can’t get access to the Supplementary Materials.
Authors respond: The supplementary material (all the data collection with the variables), were sent according to the journal regulations, they asked for a link for reviewers access, we will check it, but I have checked the link and have access, you can see in this photograph. We have removed the restrictions so you should be able to access the file.
- In the Disscusion section, page 7, line 236 you should delete one of ‘or’.
Authors respond: Thank you, done.

Reviewer 2 Report
The reviewed text tackles an interesting issue from the point of view of the functioning of older people and their care. The aim of the research is typically diagnostic and is in principle fulfilled, while the derivation of conclusions for practice, which would certainly strengthen the value of the text, is missing. The authors only conclude that future research may provide such knowledge: "Future research may use the results of this study to assess and determine the long-term effect of preventive and compulsory social isolation measures due to the Covid-19 pandemic".
In my opinion, the description of the research procedures has some shortcomings and the research results are presented too briefly and sometimes lack consistency. Here are some examples:
Line 129: "A total of 255 emails were sent, of which 89 responded and another 14 did not". This sentence in this form is unreadable, especially when juxtaposed with the next one.
Line 151-152: "Participant sociodemographic characteristics and the relationship between sociodemographic variables, the Barthel Index, and the Lawton & BrodyScale are displayed in Table 1". Meanwhile, we find no demographic data or references to the Barthel Index in the table. Demographic data only appear as scattered in a few other places: Line 150: "A total of 193 participants were included in the study with mean age of 76.56 years (121 women and 72 men)" or Line 192: "In the present study, the mean age was 76.56 with slightly more women than men"; Line 204: "These results are consistent with ours, where 40% of participants completed secondary education...". So it appears that the table with demographic data has been completely omitted from the text, which makes it very difficult to evaluate the research conducted.
Some of the findings are overly obvious and do not add any new knowledge, e.g: Line 165: "The lowest values of independence were recorded between the age ranges 90-94 years".
In addition, Spanish language wording appears in several places: Line 178: ABVD (Barthel) - instead of BADL, or figure 4.
Some sentences are not very understandable, such as: Line 26: "The activities which greatest difficulties for their execution of basic activities were going up and down stairs (22%) and moving around (18%) and the greatest difficulties for instrumental activities of daily living were shopping (22%) and preparing food (15%)"; Line 154: "Only 1% recorded lower percentages of independence, in scores between 70 and 10%, respectively".
The discussion devotes too much space to what has not been studied, e.g.: Line 193: "In this preliminary study we have not taken into consideration the cognitive impairment of the participants, which could have influenced our results as shown by previous research" - followed by a long section on 'cognitive impairment'". There is also too much focus on the falls of older people....
Author Response
Comments and Suggestions for Authors
The reviewed text tackles an interesting issue from the point of view of the functioning of older people and their care. The aim of the research is typically diagnostic and is in principle fulfilled, while the derivation of conclusions for practice, which would certainly strengthen the value of the text, is missing. The authors only conclude that future research may provide such knowledge: "Future research may use the results of this study to assess and determine the long-term effect of preventive and compulsory social isolation measures due to the Covid-19 pandemic".
Authors: Thank you very much for your dedication and for your comments that will help us improve the article. We have worked on the discussion and conclusions according to the research and the text. We have rewritten the conclusions as follows to give more value to the result of the investigation: (beginning line 283): Althoughwe cannot establish an exact correlation since the period evaluated in our study does not exceed 6 months, itis a fact that is worth studying in greater depth in order to initiate future preventive plans. However, we shouldtake into consideration, that maybe the perception of independence that participants had at the time of thesurvey could be different at the end of the pandemic and should be considered in future studies. Doing so, would support establishing prevention programs to improve and/or maintain the independence of dailyactivities and mobility. (Conclusion): COVID-19 has caused isolation leading to functional limitations for many, especially the older adult. The current study highlighted participants perceptions of decreased function in specific activities. Perceived declines in function and mobility may lead to decreased independence and safety for the older adult; therefore preventative planning and programming should be considered.
In my opinion, the description of the research procedures has some shortcomings and the research results are presented too briefly and sometimes lack consistency. Here are some examples: Line 129: "A total of 255 emails were sent, of which 89 responded and another 14 did not". This sentence in this form is unreadable, especially when juxtaposed with the next one.
Authors: This has been clarified to read as follows (line 137): A total of 255 emails were sent, of which 89 agreed to participate. One participant responded, but she did not wish to participate in the study. Of the 165 emails that were not answered, 92 participants were recruited via phone and agreed to participate in the study. An additional 12 participants who met the inclusion criteria were recruited from a randomized list from the rehabilitation service of the hospital. A total of 193 participants were included in this study.
Line 151-152: "Participant sociodemographic characteristics and the relationship between sociodemographic variables, the Barthel Index, and the Lawton & BrodyScale are displayed in Table 1". Meanwhile, we find no demographic data or references to the Barthel Index in the table.
Demographic data only appear as scattered in a few other places: Line 150: "A total of 193 participants were included in the study with mean age of 76.56 years (121 women and 72 men)" or Line 192: "In the present study, the mean age was 76.56 with slightly more women than men"; Line 204: "These results are consistent with ours, where 40% of participants completed secondary education...". So it appears that the table with demographic data has been completely omitted from the text, which makes it very difficult to evaluate the research conducted.
Authors: Table 1 in the text did not correspond with the reference. Thank you. We have changed the table.
Some of the findings are overly obvious and do not add any new knowledge, e.g: Line 165: "The lowest values of independence were recorded between the age ranges 90-94 years".
In addition, Spanish language wording appears in several places: Line 178: ABVD (Barthel) - instead of BADL, or figure 4.
Authors: Changes and corrections made.
Some sentences are not very understandable, such as: Line 26: "The activities which greatest difficulties for their execution of basic activities were going up and down stairs (22%) and moving around (18%) and the greatest difficulties for instrumental activities of daily living were shopping (22%) and preparing food (15%)"; Line 154: "Only 1% recorded lower percentages of independence, in scores between 70 and 10%, respectively".
Authors: We have clarified as follows: (line 26) The activities with the greatest difficulties included going up and down stairs (22%) and moving around (18%)… (line 172) Regarding the assessment through the Barthel Index, 73% reported being independent in their BADL, reaching the maximum score of 100. Only 1% recorded lower percentages of independence. The greatest limitations reported by participants included going up and down the stairs (22%) and difficulty in moving around (18%). Significant differences are observed between the values ​​of the Barthel Index and the age ranges (p 0.01) (Fig. 2) and between the values ​​of the Lawton & Brody Scale and the age ranges (p 0.001) (Fig. 3).
The discussion devotes too much space to what has not been studied, e.g.: Line 193: "In this preliminary study we have not taken into consideration the cognitive impairment of the participants, which could have influenced our results as shown by previous research" - followed by a long section on 'cognitive impairment'". There is also too much focus on the falls of older people....
Authors: Thank you for the observation, we have worked on discussion section. However, we made refence to the cognitive aspect because it’s an important aspect that could not be evaluated in our study due to the social isolation. We made a special mention in the text as a limitation of the study. Subsequently, we have focused on falls because it’s an aspect of crucial importance in older people, especially considering that many of these falls later lead to dependency.

Reviewer 3 Report
Generally speaking, I find the information included in this paper to be interesting. There are some difficulties related to grammar and English language usage. I do believe that I understand the initial intent of this study, but the paper does not, in my opinion, provide enough data analysis and discussion to adequately demonstrate that the original intent and goals of the research were achieved. I find the analysis and discussion to be very scant. Throughout the document, the authors point out the gaps in existing research, but have not been able to demonstrate that they have filled in those data gaps with their own research.
The authors state that this study was carried out "six months after the period of preventative closure" as a result of the pandemic. However, it is not clear (to me) whether the study was carried out six months after the shut-downs began, or 6 months after shut-downs had concluded. This is unclear, and I believe that this is important information that might very well impact on the study outcomes.
The two scales identified and used in the study are relevant and well-established.
The authors did a good job of discussing participant selection and exclusion criteria, as well as outlining the process followed.
Again - I find the paper lacking in overall in-depth discussion of data analysis. Because there is so little discussion included, I find myself questioning what actually emerged from the gathered data. The authors have not talked enough about interpretation of results.
I am also frustrated by the scant conclusion. It appears that the main conclusion drawn is that more in-depth, and seemingly "different" research is required. This leaves me wondering about the relevance of the data gathered in this particular study. I also have to admit that the lack of relevant comparison leaves me wondering if the results are comparable in any meaningful way. The way in which this paper is written currently seems to present the gathered data in a bit of a vacuum. As a reader, I am left wondering about the relevance, and the actual connection to what the authors' stated intent was.
Author Response
Comments and Suggestions for Authors
Generally speaking, I find the information included in this paper to be interesting. There are some difficulties related to grammar and English language usage. I do believe that I understand the initial intent of this study, but the paper does not, in my opinion, provide enough data analysis and discussion to adequately demonstrate that the original intent and goals of the research were achieved. I find the analysis and discussion to be very scant. Throughout the document, the authors point out the gaps in existing research, but have not been able to demonstrate that they have filled in those data gaps with their own research.
Authors: Thank you very much for your time and dedication as well as your comments that have helped us to improve this manuscript. We have worked on the entire article in order to add clarification, value and improve its wording.
The authors state that this study was carried out "six months after the period of preventative closure" as a result of the pandemic. However, it is not clear (to me) whether the study was carried out six months after the shut-downs began, or 6 months after shut-downs had concluded. This is unclear, and I believe that this is important information that might very well impact on the study outcomes.
Authors: This has been clarified to read as follows (line 94): ”A cross-sectional observational study was carried out 6 months after the mandatory isolation period caused by COVID-19 began.” (For this reason, in the conclusion we add that it is necessary to carry out additional long term studies after shut-down have concluded, because the state of independence in daily tasks may change over time.)
The two scales identified and used in the study are relevant and well-established.
The authors did a good job of discussing participant selection and exclusion criteria, as well as outlining the process followed.
Again - I find the paper lacking in overall in-depth discussion of data analysis. Because there is so little discussion included, I find myself questioning what actually emerged from the gathered data. The authors have not talked enough about interpretation of results.
Authors: Thank you, we have worked on it. We have improved the text as a whole as well as the figures and tables that help to explain it.
I am also frustrated by the scant conclusion. It appears that the main conclusion drawn is that more in-depth, and seemingly "different" research is required. This leaves me wondering about the relevance of the data gathered in this particular study. I also have to admit that the lack of relevant comparison leaves me wondering if the results are comparable in any meaningful way. The way in which this paper is written currently seems to present the gathered data in a bit of a vacuum. As a reader, I am left wondering about the relevance, and the actual connection to what the authors' stated intent was.
Authors: We have clarified conclusions as follows: “COVID-19 has caused isolation, leading to functional limitations for many, especially the older adult. The current study highlighted participants perceptions of decreased function in specific activities. Perceived declines in function and mobility may lead to limited independence and safety for the older adult, therefore preventative planning and programming should be considered.”

Round 2
Reviewer 3 Report
I do appreciate the work that the authors have put into revising this article. I remain concerned with grammatical issues and issues related to English language usage throughout. I think that the authors have (in this version) provided more clarity within the introduction, and within the discussion section of the article. I appreciate the clear statements related to the limitations of the results.